Proteomic landscape of porcine induced neural stem cell reprogramming and differentiation

Ploypetch Sekkarin 1
http://orcid.org/0000-0003-2746-5353 Phochantachinda Sataporn 1
Chakritbudsabong Warunya 2 3
Sakcamduang Walasinee 1
Chaisilp Nattarun 4
Chaiwattanarungruengpaisan Somjit 4
Pannengpetch Supitcha 5
Na Nakorn Piyada 5
Muangthong Tharathip 6
Rungarunlert Sasitorn 2 3 sasitorn.run@mahidol.ac.th
1 Department of Clinical Sciences and Public Health, Faculty of Veterinary Science, Mahidol University , Nakhon Pathom , Thailand
2 Department of Pre-Clinic and Applied Animal Science, Faculty of Veterinary Science, Mahidol University , Nakhon Pathom , Thailand
3 Laboratory of Cellular Biomedicine and Veterinary Medicine, Faculty of Veterinary Science, Mahidol University , Nakhon Pathom , Thailand
4 The Monitoring and Surveillance Center for Zoonotic Diseases in Wildlife and Exotic Animals, Faculty of Veterinary Science, Mahidol University , Nakhon Pathom , Thailand
5 Center for Research Innovation and Biomedical informatics, Faculty of Medical Technology, Mahidol University , Nakhon Pathom , Thailand
6 Department of Pathobiology, Faculty of Science, Mahidol University , Bangkok , Thailand
Beddoe Travis
Electronic publication date: 2025 Oct 3
Publication date: 2025
Volume: 13
Electronic Location ID: e20120
Received 2025 May 28; Accepted 2025 Sep 1
Copyright: © 2025 Ploypetch et al.
Copyright year: 2025
Copyright holder: Ploypetch et al.
License: This is an open access article distributed under the terms of the Creative Commons Attribution License, which permits unrestricted use, distribution, reproduction and adaptation in any medium and for any purpose provided that it is properly attributed. For attribution, the original author(s), title, publication source (PeerJ) and either DOI or URL of the article must be cited.
License URL: https://creativecommons.org/licenses/by/4.0/

Keywords: Induced neural stem cells, piNSCs, Porcine, Proteomics, Direct reprogramming, Neuronal differentiation, Mass spectrometry

Funding: Specific League Funds from Mahidol University This research is supported by Specific League Funds from Mahidol University. The funders had no role in study design, data collection and analysis, decision to publish, or preparation of the manuscript.

==============================
Background

Direct reprogramming of somatic cells into induced neural stem cells (iNSCs) holds strong potential for regenerative medicine, especially in large animal models like pigs, which are crucial for translational and preclinical research. However, the molecular mechanisms underlying porcine fibroblast-to-iNSC reprogramming and subsequent differentiation remain poorly understood at the proteomic level.

Methods

To map the proteomic landscapes associated with reprogramming and differentiation, we performed unbiased label-free discovery proteomics (nano-LC-MS/MS) and targeted SWATH-MS quantification. Proteomes of porcine tail fibroblasts (PTFs; passage 3), two porcine iNSC lines (piNSCs; VSMUi002-B and VSMUi002-E, passage 20), and their differentiated progeny (piNSCs-NGs; VSMUi002-B-NGs and VSMUi002-E-NGs, representing piNSCs at passage 20 cultured for an additional 14 days under differentiation conditions) were compared. Two previously established piNSC lines (VSMUi002-B and VSMUi002-E), generated via Sendai virus-mediated reprogramming, were used as the cellular models.

Results

The piNSC lines displayed hallmark neural stem cell (NSC) morphology and expressed canonical markers (PAX6+, SOX2+, NES+, VIM+, OCT4−). Upon differentiation, they generated neuronal and glial cells expressing TUJ1, MAP2, SYP, TH, and GFAP, confirming their multipotency. A total of 4,094 proteins were identified across the three cell states. Multivariate analysis revealed distinct proteomic signatures separating fibroblasts, iNSCs, and their neuronal/glial progeny. The proteomic shift from the fibroblast to the piNSC state was marked by increased expression of stathmin 1 (STMN1), neurofilament light polypeptide (NEFL), aconitate hydratase (ACO2), electron transfer flavoprotein subunit beta (ETFB), fructose-bisphosphate aldolase B (ALDOB), and transketolase (TKT), alongside suppression of actin-related protein 2/3 complex subunit 5 (ARPC5) and LIM domain and actin-binding protein 1 (LIMA1). These shifts indicate a dismantling of the fibroblast cytoskeleton and a broad upregulation of cellular energy and biosynthetic metabolism, reflecting a loss of fibroblast identity and the acquisition of an NSC state. Upon differentiation into piNSCs-NGs, 19 proteins were consistently upregulated. These included neuronal structural proteins (INA, STMN1), cytoskeletal regulators (PFN1), signaling modulators (MBIP), and proteins involved in lysosomal function (NCOA7), cell adhesion (CDHR2), and calcium signaling (ANXA4). Pathway and network analyses highlighted post-transcriptional regulation—particularly involving RNA processing and the RNA exosome complex (e.g., EXOSC3)—as a key feature of differentiation.

Conclusion

This study provides the first comprehensive proteomic map of piNSC reprogramming and differentiation in a large animal model. Our findings uncover critical regulatory proteins and pathways governing cytoskeletal organization, metabolism, and RNA processing, offering valuable insights into neural fate states. This resource advances the understanding of neural reprogramming in translational models and supports future regenerative and comparative neuroscience efforts.

Introduction

The directed reprogramming of somatic cells into lineage-specific progenitors, such as induced neural stem cells (iNSCs), offers considerable promise for regenerative medicine and disease modeling (Han et al., 2012; Xiao et al., 2018). Direct reprogramming avoids the pluripotent intermediate stage typical of induced pluripotent stem cell (iPSC) generation. This approach presents a potentially safer strategy, mitigating the teratoma risk associated with residual undifferentiated pluripotent cells (Meng et al., 2013), as well as a more streamlined and potentially faster route for obtaining neural precursors suitable for therapeutic development and in vitro neurological studies (Hou & Lu, 2016). Although foundational iNSC generation protocols have been established in rodent and human systems (Han et al., 2012; Ring et al., 2012; Xiao et al., 2018), translating these approaches to large animal models is crucial for preclinical validation (Lamanna et al., 2017). Among large animal models, the pig (Sus scrofa) is increasingly recognized for preclinical translation due to its significant physiological, anatomical, organ size, and genetic similarities to humans (Lunney et al., 2021), making porcine iNSCs (piNSCs) a particularly valuable platform. Building on established methods, including our previous work employing integration-free Sendai virus (SeV) systems (Chakritbudsabong et al., 2022), the generation of piNSCs from porcine fibroblasts has been successfully demonstrated. These previously established piNSC lines displayed characteristics consistent with neural stem cells (NSCs): a neuroepithelial-like morphology, neurosphere formation capability, and robust expression of key neural stem cell (NSC) markers (PAX6, SOX2, NESTIN, and VIMENTIN). Furthermore, they demonstrated proliferative capacity (Ki-67-positive) and multipotency, differentiating into neurons, astrocytes, and oligodendrocytes. Critically, the absence of pluripotency marker expression (e.g., OCT4, NANOG) confirmed successful conversion bypassing pluripotency (Chakritbudsabong et al., 2022). These well-characterized piNSCs are the subject of the current proteomic investigation.

Nevertheless, a significant gap remains in understanding the molecular mechanisms, particularly at the protein level, that govern not only the initial fibroblast-to-iNSC conversion but also the subsequent differentiation of piNSCs into mature neural lineages. Most studies to date have relied on transcriptomic analyses or key marker expression (Han et al., 2020; Xu et al., 2020), which provide an incomplete picture. The proteome, in contrast, offers a more direct reflection of the cell’s functional state, capturing critical post-transcriptional and post-translational regulatory events, protein turnover dynamics, and subcellular localization often missed by transcriptomics (Gerst, 2018). Therefore, characterizing the comprehensive proteomic alterations that define both the establishment of a stable NSC identity from somatic precursors and the subsequent commitment to differentiated neural fates is crucial for identifying key regulatory nodes, potential reprogramming bottlenecks, and markers of functional maturity. This knowledge is pivotal for optimizing the generation of functionally competent neural subtypes essential for therapeutic efficacy.

Previous proteomic investigations of porcine neural stem cells, primarily focusing on fetal NSC differentiation using techniques such as 2-DE (Skalnikova et al., 2007), have laid foundational groundwork. However, a comprehensive, high-resolution proteomic atlas detailing the molecular events at the protein level during the direct reprogramming of somatic cells into piNSCs and their subsequent lineage-specific differentiation remains to be established. The current study leverages advanced quantitative mass spectrometry to address this gap, providing novel insights into the molecular landscapes of these distinct cellular states in a key translational large animal model.

This focus on a large animal model is pertinent because, even with progress in other systems, a comprehensive understanding remains elusive. For instance, proteomic studies of human NSC differentiation and iPSC-derived NSCs have indeed highlighted significant protein alterations. These include the dynamic regulation of the Wnt/β-catenin pathway (Červenka et al., 2021; Urasawa et al., 2023), changes in proteasome subunit composition (Álvarez et al., 2023), and key phosphorylation events that govern cell fate (Shoemaker & Kornblum, 2016; Hardt et al., 2023). However, a comparable and integrative understanding of these processes during the de novo generation and subsequent maturation of directly reprogrammed iNSCs, particularly porcine piNSCs, is still missing. This study was undertaken to address these critical knowledge gaps. Our primary objectives were twofold: first, to comprehensively map the proteomic differences that define the stable piNSC state relative to the original porcine tail fibroblasts (PTFs); and second, to delineate the proteomic signatures associated with the subsequent directed differentiation of these piNSCs into diverse piNSCs-NGs. Based on findings in various NSC systems, we hypothesized that distinct and quantifiable proteomic shifts underpin both the acquisition of the piNSC state and the subsequent differentiation process. We anticipated these shifts would involve several key molecular features. These include the modulation of crucial signaling networks, such as Wnt/β-catenin, HIF-1, and VEGF (Červenka et al., 2021; Urasawa et al., 2023), and substantial metabolic reprogramming (Červenka et al., 2021; Hardt et al., 2023). Additionally, we expected to observe remodeling of the cytoskeleton (Hardt et al., 2023) and proteostasis machinery (Álvarez et al., 2023). Finally, we posited that critical post-translational modifications (PTMs), particularly phosphorylation events known to dictate protein activity and processing of key neural regulators like Dclk1 (Shoemaker & Kornblum, 2016; Hardt et al., 2023), would be evident and play significant roles. Identifying the specific proteins and inferring the activity of PTM-driven pathway dynamics in the porcine system would illuminate fundamental mechanisms governing large animal neural lineage specification and commitment, potentially revealing conserved and species-specific regulatory features crucial for advancing translational applications.

To achieve these objectives, we employed a robust quantitative proteomic workflow, integrating label-free discovery proteomics (nano-liquid chromatography tandem mass spectrometry, nano-LC-MS/MS) with targeted, high-accuracy quantification using sequential window acquisition of all theoretical mass spectra (SWATH-MS). This combined approach was applied comparatively to PTFs, independently generated piNSC lines, and piNSCs-NGs. To unravel the molecular logic embedded within these rich proteomic datasets, a comprehensive suite of bioinformatic tools was deployed. Multivariate statistics were utilized to robustly identify differential protein expression patterns defining each cellular state. Hierarchical clustering further grouped proteins based on their dynamic profiles, suggesting functional coordination. Crucially, pathway enrichment analyses contextualized these changes within known biological processes and signaling cascades, while the construction and interrogation of protein-protein interaction networks aimed to elucidate the evolving architecture of functional modules and identify key regulatory hubs governing piNSC identity and neural fate commitment.

Here, we present, to our knowledge, the first in-depth proteomic characterization of both piNSC reprogramming and differentiation in a large animal model. This study details the quantitative proteomic data, reveals key protein signatures and pathways distinguishing the different cell states (including a dismantling of the fibroblast cytoskeleton, a broad upregulation of cellular energy metabolism, and the acquisition of neural lineage markers like NEFL), and discusses the functional implications of these proteomic shifts. These findings provide crucial insights into the molecular underpinnings of neural lineage commitment in a translationally significant large animal model, offering a valuable resource for advancing regenerative neurology and comparative stem cell biology.

Materials and Methods

Ethical statement

This study was conducted in accordance with the ethical guidelines for animal research established by the Institutional Animal Care and Use Committee (IACUC) at the Faculty of Veterinary Science, Mahidol University, Thailand (approval number: MUVS-2023-11-75).

Cell culture

All cells were cultured in a humidified incubator at 37 °C supplied with 5% CO2. Culture media and supplements were obtained from Thermo Fisher Scientific (Waltham, MA, USA) and Sigma-Aldrich (St. Louis, MO, USA), unless otherwise noted.

Porcine tail fibroblast culture

Primary PTFs, which served as the parental somatic cell source for generating the piNSC lines (Chakritbudsabong et al., 2022), were also utilized in the current study for comparative proteomic analysis. The isolation and initial characterization of these PTFs are detailed in our previous work (Chakritbudsabong et al., 2022). PTFs were cultured in Dulbecco’s Modified Eagle Medium (DMEM) containing high glucose (4.5 g/L) supplemented with 10% fetal bovine serum (FBS; Hyclone, Logan, UT, USA; cat. no. SV30160), 1% GlutaMAX™, and 1% Antibiotic-Antimycotic solution (containing 10,000 units/mL of penicillin, 10,000 µg/mL of streptomycin, and 25 µg/mL of Gibco Amphotericin B). Cells were seeded in 10 cm tissue culture dishes at a density of 1.5 × 104 cells/cm2 and sub-cultured every 4 days using 0.05% Trypsin-EDTA at a 1:4 split ratio. PTFs at passage 3 were used for all experiments.

Porcine induced neural stem cell (piNSC) culture

Two piNSC lines (VSMUi002-B and VSMUi002-E), previously generated and characterized (Chakritbudsabong et al., 2022), were used. piNSCs were cultured in a 1:1 mixture of DMEM/F-12 and Neurobasal medium supplemented with 1% GlutaMAX™, 1% Antibiotic-Antimycotic solution, 1% N-2 supplement, 2% B-27 supplement, 10 ng/mL human epidermal growth factor (EGF; cat. no. PHG0311L), and 20 ng/mL human basic fibroblast growth factor (bFGF; R&D Systems, Minneapolis, MN, USA; cat. no. 233-FB-025/CF). Cells were seeded on Matrigel-coated 6 cm tissue culture dishes at a density of 1.5 × 104 cells/cm2 and passaged every 2–3 days using Versene® solution (cat. no. 15040066) at a 1:5 split ratio. piNSCs at passage 20 were used for all experiments.

Differentiation of piNSCs into neuronal and glial populations (piNSCs-NGs)

Directed neural differentiation of piNSCs was induced as described previously (Chakritbudsabong et al., 2022) with minor modifications. Briefly, piNSCs were dissociated into single cells using Versene® solution and seeded at a density of 2 × 104 cells/cm2 on Matrigel-coated 6-well or 24-well plates in neural differentiation medium (piNSC base medium lacking bFGF and EGF). The medium was replaced every 2 days for 14 days. Phase-contrast images were acquired daily to monitor cell morphology changes consistent with differentiation. At day 0 (undifferentiated piNSCs, passage 20) and day 14 (differentiated piNSCs-NGs, representing the passage 20 piNSCs after 14 days of neural induction), cells were harvested and processed for immunofluorescence staining and proteomic analysis.

Immunofluorescence staining

Cells were fixed in 4% paraformaldehyde (PFA) in phosphate-buffered saline (PBS) for 15 min at room temperature (RT) and rinsed three times with PBS. Cells were then permeabilized with 0.25% Triton X-100 in PBS for 10 min at RT and blocked with 2% bovine serum albumin (BSA) in PBS for 1 h at RT. Primary antibodies (Table 1) were diluted in blocking buffer and applied to the samples, followed by incubation overnight at 4 °C. After washing three times with PBS, samples were incubated with fluorophore-conjugated secondary antibodies (Table 1) diluted in blocking buffer for 1 h at RT in the dark. Following three additional washes with PBS, slides were mounted using VECTASHIELD® Mounting Medium with DAPI (Vector Laboratories, Newark, CA, USA) and imaged using a DMi8 inverted fluorescence microscope equipped with a DFC7000 camera (Leica Microsystems, Wetzlar, Germany).

Table 1 List of antibodies.

Markers	Antibodies	Dilution	Companies	Cat #	
Pluripotency	Goat anti-OCT4	1:200	Santa Cruz Biotechnology	sc-8628	
Neural stem cells	Rabbit anti-PAX6	1:100	Thermo Fisher Scientific	PA5-85374	
	Rabbit anti-SOX2	1:100	Santa Cruz Biotechnology	sc-20088	
	Mouse anti-NESTIN	1:100	Millipore	MAB5326	
	Mouse anti-VIMENTIN	1:100	Santa Cruz Biotechnology	sc-6260	
Neural lineage	Rabbit anti-TUJ	1:100	Abcam	AB18207	
	Mouse anti-MAP	1:100	Santa Cruz Biotechnology	sc-32791	
	Rabbit anti-GFAP	1:100	Millipore	AB5804	
	Mouse anti-SYP	1:100	Abcam	AB8049	
	Rabbit anti-TH	1:100	Abcam	AB137869	
Secondary antibodies	Alexa Fluor 488 donkey anti-rabbit IgG	1:1,000	Thermo Fisher Scientific	A-21206	
Alexa Fluor 488 donkey anti-mouse IgG	1:1,000	Abcam	Ab150105	
Alexa Fluor 488 donkey anti-goat IgG	1:1,000	Thermo Fisher Scientific	A11055	
Alexa Fluor 594 donkey anti-rabbit IgG	1:1,000	Thermo Fisher Scientific	A-21207	
	Alexa Fluor 594 donkey anti-mouse IgG	1:1,000	Thermo Fisher Scientific	A-21203	

In-solution protein digestion

Protein samples (50 μg per sample) were resuspended in 100 mM triethylammonium bicarbonate (TEAB, pH 8.0) and reduced by adding 100 mM dithiothreitol (DTT) for 30 min at 37 °C. Alkylation was performed by adding 100 mM iodoacetamide (IAA) in 100 mM TEAB for 30 min at RT in the dark. Samples were then reduced again by adding 100 mM DTT for 15 min at RT. Protein digestion was carried out using MS-grade Trypsin Gold (Promega, Madison, WI, USA) at 37 °C for 16 h. Peptides were desalted and concentrated using C18 ZipTips (MilliporeSigma, Burlington, MA, USA) according to the manufacturer’s instructions and stored at −80 °C until analysis. Prior to analysis, peptides were reconstituted in 0.1% formic acid (FA), and peptide concentrations were determined using a NanoDrop 1000 spectrophotometer (Thermo Fisher Scientific (Bremen) GmbH, Bremen, Germany).

Label-free nano-LC-MS/MS analysis (DDA)

Peptides (1 μg) were analyzed using a nano-liquid chromatograph (Dionex Ultimate 3000 RSLCnano System; Thermo Fisher Scientific) coupled to a Bruker compact QTOF mass spectrometer via a CaptiveSpray source (Bruker, Bremen, Germany). Peptides were first loaded onto a C18 Nano trap column (Thermo Fisher Scientific, Waltham, MA, USA) and then separated on an analytical C18 HPLC column (Thermo Fisher Scientific, Waltham, MA, USA) at a flow rate of 300 nL/min and a column temperature of 60 °C. A linear gradient was employed using mobile phase A (0.1% FA in LC-MS grade water) and mobile phase B (0.08% FA in 80% LC-MS grade acetonitrile). The drying gas flow rate was set to 5 L/min at 150 °C. MS (MS1) data were acquired at 6 Hz in positive ionization mode, over a mass range of m/z 150 to 2,200. Automatic MS/MS fragmentation (AutoMSn) was performed using data-dependent acquisition (DDA), targeting the two most intense precursor ions, with fragmentation frequencies of 4 Hz (low intensity precursors) and 16 Hz (high intensity precursors). A dynamic exclusion time of 3 s was used to prevent re-analysis of the same ions per MS1 scan. Sodium formate clusters were used for internal mass calibration and introduced automatically via a syringe pump at the beginning of each run.

Targeted label-free nano-LC QTOF using SWATH analysis (DIA)

For SWATH-MS analysis, 1 μg of peptides per sample was analyzed using the same nano-LC-MS system and chromatographic conditions described above (including columns, mobile phases, flow rate, temperature, and gradient). Data-independent acquisition (DIA) was performed in SWATH mode. The instrument was set to acquire data over the mass range m/z 150 to 2,200 using 32 overlapping isolation windows of 25 Da width. MS/MS (MS2) fragment ion spectra were acquired for all ions within each window with an accumulation time of 100 ms per window, resulting in a total duty cycle of approximately 3.2 s (32 windows × 100 ms/window). Quadrupole resolution during precursor isolation was set to 25 Da/mass selection.

Bioinformatics and data analysis

Raw MS (DDA) files were processed using MaxQuant software (version 1.6.2.10). MS/MS spectra were searched using the integrated Andromeda search engine against the Sus scrofa UniProt database, incorporating variable modifications (methionine oxidation, N-terminal acetylation) and a fixed modification (carbamidomethyl cysteine). For the Bruker QTOF data, peptide mass tolerance was set to 0.5 Da for initial searches and 0.25 Da for main searches. Trypsin/P was specified as the enzyme, with a maximum of two missed cleavages allowed. Protein identifications adhered to a 1% false discovery rate (FDR) threshold. Label-free quantification (LFQ) was performed with a MS/MS match tolerance of 0.5 Da. Mass recalibration and retention time alignment between runs were enabled using the “Match between runs” feature in MaxQuant. LFQ data were then imported into Perseus software (version 1.6.8.0) for differential expression analysis using one-way ANOVA and post hoc Tukey’s HSD test, with an FDR threshold of 0.05. For SWATH data extraction, Skyline software (version 19.1.0.193) was used. Peak integration was manually reviewed for accuracy, and peptide intensity was determined by summing all transition peak areas (total area sum, TAS method).

Multivariate and statistical analysis

Multivariate analysis, including heatmap analysis, partial least squares-discriminant analysis (PLS-DA) plots, and dendrograms, was performed using MetaboAnalyst 6.0 (http://www.metaboanalyst.ca) (Ewald et al., 2024). Gene Ontology (GO) enrichment and Kyoto Encyclopedia of Genes and Genomes (KEGG) pathway analyses were conducted using ShinyGO 0.80 (http://bioinformatics.sdstate.edu/go80/) (Ge et al., 2020) to identify significantly enriched terms and pathways (FDR < 0.05). One-way analysis of variance (ANOVA) with post hoc Tukey’s test (p < 0.05) was used to identify differentially expressed proteins between groups. Venn diagrams were generated to visualize protein expression overlap. Protein sequences and biological processes were annotated using UniProtKB/Swiss-Prot (The UniProt Consortium, 2023). Protein-protein interactions were investigated using STITCH (http://stitch.embl.de; accessed on July 28, 2024) (Szklarczyk et al., 2016), in conjunction with NSC markers (PAX6, SOX2, NESTIN, and VIMENTIN) and neural markers.

Results

Characterization of piNSCs and their neuronal and glial differentiation

We began by confirming the NSC characteristics of the two piNSC lines, VSMUi002-B and VSMUi002-E, used in this study. These lines were previously established via Sendai viral transduction of PTFs with human OSKM factors (Chakritbudsabong et al., 2022). Consistent with their NSC identity, both cell lines, when cultured in piNSC medium as adherent monolayers, exhibited a characteristic neuroepithelial morphology, a hallmark of NSCs (Fig. 1A, top row). Immunofluorescence analysis confirmed robust expression of key NSC markers: PAX6, a crucial transcription factor for neural development; SOX2, a marker essential for pluripotency and NSC maintenance; NESTIN (NES), an intermediate filament protein characteristic of neural progenitors; and VIMENTIN (VIM), another intermediate filament protein present in various progenitor cells, including those of neural origin. Critically, the absence of the pluripotency marker OCT4 confirmed that these cells have exited the pluripotent state and are committed to a neural lineage (Fig. 1A). Both piNSC lines exhibited similar marker profiles and differentiation capacities.

Figure 1 Characterization of porcine induced neural stem cells (piNSCs) and their differentiation into neuronal and glial cells (piNSCs-NGs).

(A) Immunofluorescence of piNSC lines VSMUi-002B (representative images, left) and VSMUi-002E (representative images, right) in monolayer culture. Cells express neural stem cell markers PAX6 (green), SOX2 (red), NESTIN (NES; green), and VIMENTIN (VIM; green). Co-staining for the pluripotency marker OCT4 (green) demonstrates its absence. Nuclei are stained with DAPI (blue). (B) Immunofluorescence of differentiated piNSCs-NGs derived from VSMUi-002B (representative images, left) and VSMUi-002E (representative images, right). Cells display neuronal and glial morphologies and express neuronal markers TUJ1 (green), MAP2 (red), synaptophysin (SYP; red), and tyrosine hydroxylase (TH; red), as well as the astrocyte marker GFAP (green). For panels showing MAP2 or SYP, these markers (red) are shown with TUJ1 (green) to illustrate co-expression or distinct localization where applicable. Nuclei: DAPI (blue).

To assess their differentiation potential, piNSCs were transitioned to neural differentiation medium for 14 days. This resulted in significant morphological changes, including the development of extensive neurite outgrowth and the formation of clusters resembling neuronal networks (Fig. 1B, top row). These differentiated cells, operationally termed piNSC-derived neuronal and glial cells (piNSCs-NGs), were then analyzed for expression of neuronal and glial markers. Immunostaining of these piNSCs-NGs revealed the presence of both the immature neuronal marker neuron-specific class III beta-tubulin (TUJ1/TUBB3) and the mature neuronal marker microtubule-associated protein 2 (MAP2), suggesting the presence of neurons at different stages of maturation. Furthermore, the expression of synaptophysin (SYP), a synaptic vesicle protein, indicates the potential for synapse formation and functional neuronal activity. The presence of tyrosine hydroxylase (TH), a key enzyme in dopamine synthesis, suggests the differentiation of a dopaminergic neuronal subpopulation. Finally, the detection of glial fibrillary acidic protein (GFAP), an astrocyte marker, demonstrates the generation of glial cells alongside neurons (Fig. 1B).

Proteomic analysis of PTFs, piNSCs, and piNSCs-NGs

To define the proteomic landscape associated with the reprogramming of PTFs into piNSCs and their subsequent differentiation into neuronal and glial lineages, a comprehensive label-free quantitative (LFQ) proteomic analysis was performed. The study compared PTFs (passage 3), two piNSC lines (VSMUi002-B and VSMUi002-E, passage 20), and their differentiated progeny (piNSCs-NGs: VSMUi002-B-NGs and VSMUi002-E-NGs, representing piNSCs at passage 20 cultured for an additional 14 days under differentiation conditions). Three independent biological replicates were analyzed per group. Cell lysates from each replicate underwent in-solution trypsin digestion followed by discovery-mode nano-liquid chromatography tandem mass spectrometry (nano-LC-MS/MS) using data-dependent acquisition (DDA) (Fig. 2). Analysis of the DDA data from all individual replicates identified a total of 4,094 proteins across all samples (Table S1). Samples were processed and analyzed individually without pooling.

Figure 2 Overview of the proteomic analysis workflow for PTFs, piNSCs, and piNSCs-NGs.

(Column i) Cell culture and sample preparation: PTFs, piNSCs, and piNSCs-NGs were cultured, lysed, and proteins digested with trypsin. (Column ii) Discovery proteomics and label-free quantification (LFQ): Resulting peptides from all biological replicates were analyzed by nano-LC QTOF mass spectrometry using data-dependent acquisition (DDA). Raw DDA data were processed (e.g., using MaxQuant and Perseus) for protein identification, quantification, and initial downstream analyses including differential expression, clustering, and pathway enrichment to identify interesting proteins. (Column iii) Targeted quantitative confirmation: Selected differentially expressed or interesting proteins identified from the DDA-LFQ discovery phase were then subjected to targeted quantitative confirmation using SWATH-MS analysis. Further analysis of these confirmed candidate proteins involved assessing their biological function and involvement in protein-protein interaction networks and pathways. Image created with BioRender (biorender.com).

Multivariate analyses were employed to visualize the overall patterns of protein expression across the different cell types (PTFs, piNSCs, and piNSCs-NGs). A heatmap illustrating relative protein abundance was generated (Fig. 3A). The heatmap revealed distinct proteomic signatures for each group, visually confirming substantial differences in protein abundance between fibroblasts, piNSCs, and their differentiated progeny. To further quantify these differences and identify the major sources of variation in the proteomic data, supervised partial least squares discriminant analysis (PLS-DA) (Fig. 3B) and unsupervised hierarchical clustering analysis (Fig. 3C) were performed. The PLS-DA scores plot (Fig. 3B) demonstrated clear separation between the three main cell types (PTFs, piNSCs, and piNSCs-NGs), indicating that reprogramming and differentiation are associated with significant and distinct proteomic remodeling. Within the piNSC group, the two lines (VSMUi002-B and VSMUi002-E) clustered relatively closely, suggesting a high degree of proteomic similarity. Similarly, the differentiated cells (VSMUi002-B-NGs and VSMUi002-E-NGs) also clustered together, separated from both the PTFs and piNSCs. However, some separation was observed between the two piNSC lines and between the two piNSC-NG groups, suggesting subtle but potentially important line-specific differences. The hierarchical clustering analysis (Fig. 3C) corroborated these findings, showing a branching pattern that reflects the relationships between the cell types and lines. To explore the overall biological processes represented within the identified proteome, the complete set of 4,094 proteins identified across all samples underwent Gene Ontology (GO) enrichment analysis and Kyoto Encyclopedia of Genes and Genomes (KEGG) pathway analysis (FDR < 0.05; Table S2). Figure 3D presents the top 20 enriched GO terms and KEGG pathways. Prominent among these were terms related to “ribosome,” “regulation of mRNA metabolic process,” and “mRNA metabolic process.” This enrichment suggests a significant role for translational regulation and RNA metabolism in establishing the reprogrammed and differentiated states.

Figure 3 Comparative proteomic analysis of PTFs, piNSCs, and piNSCs-NGs.

(A) Heatmap of differentially expressed proteins across cell types: PTFs, piNSC lines (VSMUi002-B, VSMUi002-E), and derivatives (VSMUi002-B-NGs, VSMUi002-E-NGs). Rows represent proteins, and columns represent individual biological replicates (n = 3 per group). Proteins and samples are clustered using hierarchical clustering. Color intensity represents z-score normalized relative protein abundance (blue: low, red: high). (B) PLS-DA scores plot. Points: biological replicates; colors: cell type/line [PTFs (red), VSMUi002-B (green), VSMUi002-E (blue), VSMUi002-B-NGs (light blue), VSMUi002-E-NGs (purple)]. X-axis: Component 1 (4 6.9%); Y-axis: Component 2 (4.2%). (C) Dendrogram showing sample relationships. Branch lengths: dissimilarity. (D) Top 20 enriched GO/KEGG terms. X-axis: fold enrichment; Y-axis: term; circle size: gene/protein count; circle color: −log10(FDR).

Proteomic comparison of PTFs and piNSCs

To specifically characterize proteomic changes differences between PTFs and piNSCs, the LFQ protein expression profiles of PTFs and the two piNSC lines were compared. From the total of 4,094 proteins previously identified across all experimental conditions (including PTFs, piNSCs, and piNSCs-NGs), a Venn diagram analysis (Fig. 4) was constructed to visualize protein distribution among PTFs, VSMUi002-B, and VSMUi002-E. This analysis revealed that 521 proteins (12.7% of the total 4,094 identified) were detected in all three of these specific groups, representing a potential core proteome common to both PTFs and the resulting piNSCs. The two piNSC lines exclusively shared a larger set of proteins with each other (330; 8.1% of the total 4,094 identified) than either line did individually with PTFs (PTFs and VSMUi002-B shared 112 proteins, 2.7%; PTFs and VSMUi002-E shared 143 proteins, 3.5%, excluding proteins also present in the third group). Conversely, unique proteins were identified for each group relative to this three-way comparison: PTFs (695 proteins, 17.0%), VSMUi002-B (686 proteins, 16.8%), and VSMUi002-E (708 proteins, 17.3%). These observations indicate that while a core set of proteins is maintained, reprogramming induces a substantial shift in the proteome, with the piNSC lines exhibiting greater proteomic similarity to each other than to the parental PTFs.

Figure 4 Venn diagram of protein identifications in PTFs and two piNSC lines (VSMUi002-B and VSMUi002-E).

The diagram illustrates the distribution of proteins detected within PTFs (red circle), VSMUi002-B piNSCs (green circle), and VSMUi002-E piNSCs (blue circle), highlighting numbers of shared and group-specific proteins. Non-overlapping regions indicate proteins detected uniquely in that specific cell type within this comparison, while overlapping regions represent proteins shared between the indicated cell types. The central overlapping region shows 521 proteins detected in all three types. The orange outline highlights the 330 proteins detected in both piNSC lines (VSMUi002-B and VSMUi002-E) but not in PTFs, representing a key set of proteins commonly associated with the piNSC state following reprogramming from PTFs.

To identify proteins with significantly different abundance levels between PTFs and piNSCs, we analyzed the LFQ intensities of the 521 proteins detected in all three groups (PTFs, VSMUi002-B, and VSMUi002-E). One-way ANOVA with a Tukey’s post-hoc test (p < 0.05) identified 194 proteins with statistically significant differences in abundance between PTFs and at least one piNSC line (VSMUi002-B or VSMUi002-E) (Table S3). These 194 differentially expressed proteins represent potential key regulators or markers of the reprogrammed state.

SWATH-MS analysis identifies key proteins distinguishing the piNSC state from PTFs

To precisely identify and quantify key protein that distinguish the piNSC state from parental PTFs, SWATH-MS, a data-independent acquisition (DIA) method noted for its comprehensive and consistent quantification capabilities (Ludwig et al., 2018), was employed. A candidate pool for targeted SWATH-MS analysis was formed from two main protein groups: 194 proteins differentially expressed between PTFs and piNSCs (identified by ANOVA from 521 commonly detected proteins), and 330 proteins found exclusively in both piNSC lines compared to PTFs (identified by Venn diagram analysis). From this combined pool, a refined subset of 26 proteins was selected based on criteria including association with neural lineage and evidence of significant changes in expression during reprogramming (Table S4).

This targeted analysis identified eight proteins with statistically significant changes in abundance between PTFs and piNSCs (one-way ANOVA with Tukey’s HSD post-hoc test, p < 0.05). Six of these proteins were upregulated: neurofilament light polypeptide (NEFL), aconitate hydratase (ACO2), electron transfer flavoprotein subunit beta (ETFB), stathmin 1 (STMN1), fructose-bisphosphate aldolase B (ALDOB), and transketolase (TKT). The remaining two proteins were downregulated: actin-related protein 2/3 complex subunit 5 (ARPC5) and LIM domain and actin-binding protein 1 (LIMA1). Table 2 summarizes the UniProtKB/Swiss-Prot annotations for these proteins, including their biological processes, cellular components, and molecular functions.

Table 2 Nominated proteins in PTFs, VSMUi002-B, and VSMUi002-E based on biological process, cellular components, and molecular functions involvement using UniProtKB/Swiss-Prot.

Protein IDs	Protein names	Gene Names	−log10(p)	FDR	Biological process	Cellular component	Molecular function	
A0A4X1VDU1	Neurofilament light polypeptide (Neurofilament triplet L protein)	NEFL	5.177	0.00017961	Neurofilament bundle assembly [GO:0033693]	Axon [GO:0030424]; Cytoplasm [GO:0005737]; Intermediate filament [GO:0005882]; Postsynaptic intermediate filament cytoskeleton [GO:0099160]	Structural constituent of postsynaptic intermediate filament cytoskeleton [GO:0099184]	
B5APV0	Actin-related protein 2/3 complex subunit 5	ARPC5	3.7135	0.0026109	Arp2/3 complex-mediated actin nucleation [GO:0034314]; Cell migration [GO:0016477]; Regulation of actin filament polymerization [GO:0030833]	Arp2/3 protein complex [GO:0005885]; Cell projection [GO:0042995]; Cytoplasm [GO:0005737]; Nucleus [GO:0005634]	Actin filament binding [GO:0051015]	
A0A4X1W359	Aconitate hydratase, mitochondrial (Aconitase) (EC 4.2.1.3)	ACO2	3.4066	0.0035286	Tricarboxylic acid cycle [GO:0006099]	Cytosol [GO:0005829]; mitochondrion [GO:0005739]	4 iron, 4 sulfur cluster binding [GO:0051539]; Aconitate hydratase activity [GO:0003994]; Metal ion binding [GO:0046872]	
A0A5G2R0H0	Electron transfer flavoprotein subunit beta (Beta-ETF)	ETFB	3.0561	0.0059319		Mitochondrial matrix [GO:0005759]	Electron transfer activity [GO:0009055]	
Q6DUB7	Stathmin	STMN1	2.8368	0.0078636	Microtubule depolymerization [GO:0007019]; neuron projection development [GO:0031175]; regulation of microtubule polymerization or depolymerization [GO:0031110]	Cytoplasm [GO:0005737]; microtubule [GO:0005874]; neuron projection [GO:0043005]	Tubulin binding [GO:0015631]	
A0A286ZYX8	Fructose-bisphosphate aldolase (EC 4.1.2.13)	ALDOA	2.4907	0.014538	Glycolytic process [GO:0006096]	I band [GO:0031674]; M band [GO:0031430]	Fructose-bisphosphate aldolase activity [GO:0004332]	
B0KYV5	LIM domain and actin-binding protein 1 (Epithelial protein lost in neoplasm)	LIMA1 EPLIN	2.1105	0.029903	Actin filament bundle assembly [GO:0051017]; cell migration [GO:0016477]; cholesterol homeostasis [GO:0042632]; cholesterol metabolic process [GO:0008203]; intestinal cholesterol absorption [GO:0030299]; ruffle organization [GO:0031529]	actin cytoskeleton [GO:0015629]; brush border membrane [GO:0031526]; cleavage furrow [GO:0032154]; cytoplasm [GO:0005737]; focal adhesion [GO:0005925]; plasma membrane [GO:0005886]; ruffle [GO:0001726]	Actin filament binding [GO:0051015]; metal ion binding [GO:0046872]	
A8U4R4	Transketolase	TKT	1.8612	0.046461	Glyceraldehyde-3-phosphate biosynthetic process
pentose-phosphate shunt
regulation of growth
xylulose 5-phosphate biosynthetic process	Cytosol
nuclear body	Calcium ion binding
magnesium ion binding
protein homodimerization activity
transketolase activity	

Protein-protein interaction network analysis of the reprogrammed piNSC state

To elucidate the molecular mechanisms and potential regulatory interactions underlying the reprogrammed piNSC state, a protein-protein interaction (PPI) network was constructed (Fig. 5). This network incorporated the eight proteins identified as significantly differentially expressed in the SWATH analysis (NEFL, ARPC5, ACO2, ETFB, STMN1, ALDOB, LIMA1, and TKT) and four established NSC markers (PAX6, SOX2, NES, and VIM).

Figure 5 The network displays interactions among the eight proteins identified as significantly differentially expressed by SWATH analysis (orange circles; NEFL, ARPC5, ACO2, ETFB, STMN1, ALDOB, LIMA1, and TKT), four established neural stem cell (NSC) markers (green).

Nodes represent proteins; edges represent predicted protein-protein interactions. The network was generated using the STITCH database (accessed July 28, 2024), integrating evidence from experimental data, text mining, and curated databases. Edge thickness represents the confidence score of the interaction. Node size is proportional to the degree of connectivity. Proteins are grouped by key KEGG pathways using colored highlights: downregulated proteins ARPC5 and LIMA1 are involved in Arp2/3 complex-mediated actin nucleation (light blue area), while the upregulated proteins ACO2, ALDOB, and TKT participate in carbon metabolism (light orange area) and the pentose phosphate pathway (light green area).

The resulting network (Fig. 5) reveals a complex interplay between the differentially expressed proteins and known NSC markers. Several of the SWATH-identified proteins (ARPC5, ACO2, ETFB, ALDOB, LIMA1, and TKT) show multiple interactions within the network, including connections to the NSC markers PAX6, SOX2, and NES. These interactions suggest the involvement of these differentially expressed proteins in established NSC regulatory pathways. NEFL, a neuronal intermediate filament protein, shows a strong interaction with VIM (vimentin), another intermediate filament protein, consistent with their known roles in cytoskeletal structure, and providing a potential link between NEFL upregulation and the morphological changes observed during reprogramming. In contrast, STMN1 exhibits limited connectivity within the network, which may reflect its known role as a microtubule-destabilizing protein, a function that might be less directly integrated into core transcriptional regulatory networks. The ARP2/3 complex proteins (ARPC5, ARPC1A, ARPC2, ARPC3, ARPC4, ACTR2, ACTR3) show extensive interconnectivity, reflecting their known association within the ARP2/3 complex.

To provide functional context, the network was enriched with KEGG pathway analysis, visually grouping proteins into key functional modules (Fig. 5). This analysis highlights two major biological themes underlying the reprogrammed state. A distinct module of downregulated proteins, including ARPC5 and LIMA1, is involved in “Arp2/3 complex-mediated actin nucleation” (light blue area), signifying a dismantling of the fibroblast cytoskeleton. A more detailed enrichment analysis (Table S5) further supports this, identifying other significantly enriched pathways such as “cytoskeleton structure” and “RHO GTPases Activate WASPs and WAVEs.” Concurrently, a large module of upregulated proteins involved in “carbon metabolism” (light orange area) and the “pentose phosphate pathway” (light green area) includes ACO2, ALDOB, and TKT. The enrichment of these distinct cytoskeletal and metabolic pathways directly illustrates the profound remodeling that characterizes the shift from a fibroblast to a NSC state.

Proteomic analysis of piNSCs and piNSCs-NG differentiation

To investigate the proteomic differences between piNSCs and their differentiated neuronal and glial progeny (piNSCs-NGs), protein expression profiles of two piNSC lines (VSMUi002-B and VSMUi002-E) and their differentiated progeny (VSMUi002-B-NGs and VSMUi002-E-NGs) were compared. A Venn diagram (Fig. 6) depicts protein overlap and uniqueness.

Figure 6 Venn diagram of protein identifications in piNSCs and piNSCs-NGs.

The diagram shows the number of proteins detected in two porcine induced neural stem cell lines (VSMUi002-B: green; VSMUi002-E: blue) and their corresponding differentiated neuronal and glial derivatives (VSMUi002-B-NGs: red; VSMUi002-E-NGs: yellow). Numbers in non-overlapping regions indicate proteins unique to a group. Overlapping regions indicate proteins shared between/among groups. The central overlap represents 426 proteins detected in all four groups. The overlap between only the two piNSC-NG groups (69 proteins) highlights proteins potentially upregulated or newly expressed during differentiation.

A core set of 426 proteins (10.4% of the total 4,094 identified proteins) was detected in all four groups (both piNSC lines and both piNSC-NG lines), suggesting that these proteins are fundamental to both the undifferentiated NSC state and the differentiated neuronal/glial state. Interestingly, 69 proteins (1.7%) were detected only in the two piNSC-NG groups (VSMUi002-B-NGs and VSMUi002-E-NGs) and not in either of the undifferentiated piNSC lines. This suggests that these 69 proteins may be specifically upregulated or newly expressed in the differentiated neuronal/glial cells. Unique proteins were also identified for each cell type: VSMUi002-B (572 proteins, 13.97%), VSMUi002-E (530 proteins, 12.95%), VSMUi002-B-NGs (536 proteins, 13.09%), and VSMUi002-E-NGs (426 proteins, 10.41%). Protein sharing was also observed: VSMUi002-B and VSMUi002-E (237 proteins, 5.79%); VSMUi002-B-NGs and VSMUi002-E-NGs (124 proteins, 3.03%); VSMUi002-B and VSMUi002-B-NGs (220 proteins, 5.37%); VSMUi002-B and VSMUi002-E-NGs (134 proteins, 3.27%); VSMUi002-E and VSMUi002-B-NGs (38 proteins, 0.93%); VSMUi002-E and VSMUi002-E-NGs (68 proteins, 1.66%).

To identify proteins with statistically significant changes in abundance between the undifferentiated and differentiated states, we focused on two sets of proteins: the 426 proteins common to all four groups and the 69 proteins unique to the piNSCs-NGs. One-way ANOVA (p < 0.001) followed by a post-hoc Tukey’s test (p < 0.05) identified 28 proteins with significant expression changes across all four groups and 22 proteins with significant changes specifically between the two piNSC-NG groups (Table S6). These differentially expressed proteins are potential key regulators and markers of the differentiation process.

Further supporting the proteomic differences, partial least squares discriminant analysis (PLS-DA) revealed distinct clustering of the four groups (Figs. S1A and S1B). Hierarchical clustering analysis, visualized as heatmaps, highlighted differentially expressed proteins across all four groups and specifically within the piNSC-NGs (Figs. S1C and S1D).

SWATH analysis identifies proteins upregulated in differentiated piNSCs-NGs

To identify potential protein markers associated with the differentiation of piNSCs into neuronal and glial lineages (piNSCs-NGs), label-free nanoLC-QTOF analysis with SWATH-MS was performed. Initially, a comparison of VSMUi002-E-NGs to undifferentiated VSMUi002-E cells identified 30 proteins with significant abundance changes. Of these 30 proteins, further analysis revealed that 28 proteins showed significant changes across all four groups (VSMUi002-B, VSMUi002-E, VSMUi002-B-NGs, and VSMUi002-E-NGs). Within this set of 28 proteins, 22 proteins also showed significant differences specifically between the two piNSC-NG groups (VSMUi002-B-NGs vs. VSMUi002-E-NGs). These 30 proteins, identified as potentially important for differentiation, were selected for further targeted analysis to quantify their expression levels in individual samples. One-way ANOVA (p < 0.05) followed by Tukey’s post-hoc test (p < 0.05) was used to determine statistically significant differences in the expression of these 30 proteins across the four groups (VSMUi002-B, VSMUi002-E, VSMUi002-B-NGs, and VSMUi002-E-NGs) (Table S7).

Targeted SWATH analysis revealed 19 proteins with significantly higher expression levels in both piNSC-NG groups (VSMUi002-B-NGs and VSMUi002-E-NGs) compared to the undifferentiated piNSC groups (VSMUi002-B and VSMUi002-E): alpha-internexin (INA), potassium calcium-activated channel subfamily M regulatory beta subunit 3 (KCNMB3), UPAR/Ly6 domain-containing protein (ENSGGOG00000001927), annexin A4 (ANXA4), exosome component 3 (EXOSC3), histone H1.3 (HIST1H1D), a J domain-containing protein, zinc finger protein (ZFPL1), nuclear receptor corepressor 2 (NCOR2), an H15 domain-containing protein, profilin 1 (PFN1), calcium-binding mitochondrial carrier protein SCaMC-3 (SLC25A23), a phospholipase B-like, cadherin related family member 2 (CDHR2), maspardin (SPG21), MAP3K12 binding inhibitory protein 1 (MBIP), transmembrane protein 263 (TMEM263), STMN1 and nuclear receptor coactivator 7 (NCOA7). Table 3 provides UniProtKB/Swiss-Prot annotations for these proteins. The consistent upregulation of these 19 proteins in the differentiated cells, across both independent lines, strongly suggests their involvement in neuronal and glial differentiation.

Table 3 Nominated proteins in VSMUi002-B, VSMUi002-E, VSMUi002-B-NGs, and VSMUi002-E-NGs based on biological process, cellular components, and molecular functions involvement using UniProtKB/Swiss-Prot.

Protein IDs	Protein names	Gene Names	−log10(p)	FDR	Biological process	Cellular component	Molecular function	
F1S847_PIG	Alpha-internexin (Internexin neuronal intermediate filament protein alpha)	INA	2.1044	0.0094362	Cellular response to leukemia inhibitory factor [GO:1990830]; intermediate filament organization [GO:0045109]; neurofilament cytoskeleton organization [GO:0060052]; postsynaptic modulation of chemical synaptic transmission [GO:0099170]	Cytoplasm [GO:0005737]; cytoplasmic ribonucleoprotein granule [GO:0036464]; intermediate filament [GO:0005882]; neurofilament [GO:0005883]; postsynaptic intermediate filament cytoskeleton [GO:0099160]; Schaffer collateral-CA1 synapse [GO:0098685]	Structural constituent of postsynaptic intermediate filament cytoskeleton [GO:0099184]	
I3LU62_PIG	Calcium-activated potassium channel subunit beta (BKbeta) (BK channel subunit beta) (Calcium-activated potassium channel, subfamily M subunit beta) (Charybdotoxin receptor subunit beta) (K(VCA)beta) (Maxi K channel subunit beta) (Slo-beta)	KCNMB3	1.484	0.035152	Detection of calcium ion [GO:0005513]; neuronal action potential [GO:0019228]	Voltage-gated potassium channel complex [GO:0008076]	Calcium-activated potassium channel activity [GO:0015269]; potassium channel regulator activity [GO:0015459]	
A0A4X1W3F4_PIG	LY6/PLAUR domain containing 4	LYPD4	3.7092	0.0003447		Plasma membrane raft [GO:0044853]		
ANXA4_PIG	Annexin A4 (35-beta calcimedin) (Annexin IV) (Annexin-4) (Chromobindin-4) (Endonexin I) (Lipocortin IV) (P32.5) (PP4-X) (Placental anticoagulant protein II) (PAP-II) (Protein II)	ANXA4 ANX4	10.086	3.803E−10		Cytoplasm [GO:0005737]; nucleus [GO:0005634]; plasma membrane [GO:0005886]; vesicle membrane [GO:0012506]; zymogen granule membrane [GO:0042589]	Calcium ion binding [GO:0005509]; calcium-dependent phospholipid binding [GO:0005544]; phosphatidylserine binding [GO:0001786]	
A0A5G2QKW4_PIG	Exosome component 3	EXOSC3	1.8667	0.015104		Exosome (RNase complex) [GO:0000178]; nucleolus [GO:0005730]	RNA binding [GO:0003723]	
A0A480TWI8_PIG	Histone H1.3		6.7049	4.5527E−07	Nucleosome assembly [GO:0006334]	Nucleosome [GO:0000786]; nucleus [GO:0005634]	DNA binding [GO:0003677]; structural constituent of chromatin [GO:0030527]	
F1SM19_PIG	J domain-containing protein	DNAJB3	5.688	4.3954E−06	Chaperone-mediated protein folding [GO:0061077]	Cytoplasm [GO:0005737]; nucleus [GO:0005634]	Hsp70 protein binding [GO:0030544]; protein folding chaperone [GO:0044183]; protein-folding chaperone binding [GO:0051087]; unfolded protein binding [GO:0051082]	
Q29294_PIG	deleted		26.212	9.2034E−26				
F1RFN3_PIG	Nuclear receptor corepressor 2	NCOR2	49.935	3.4873E−49	Negative regulation of androgen receptor signaling pathway [GO:0060766]; negative regulation of miRNA transcription [GO:1902894]; negative regulation of transcription by RNA polymerase II [GO:0000122]; regulation of cellular ketone metabolic process [GO:0010565]	Chromatin [GO:0000785]; nuclear body [GO:0016604]; nuclear matrix [GO:0016363]; transcription repressor complex [GO:0017053]	DNA binding [GO:0003677]; histone deacetylase binding [GO:0042826]; Notch binding [GO:0005112]; transcription corepressor activity [GO:0003714]	
A0A4X1TWR8_PIG	Histone H1x	LOC100516295	7.1953	1.7396E−07	Chromosome condensation [GO:0030261]; negative regulation of DNA recombination [GO:0045910]; nucleosome assembly [GO:0006334]	Nucleolus [GO:0005730]; nucleoplasm [GO:0005654]; nucleosome [GO:0000786]	Double-stranded DNA binding [GO:0003690]; nucleosomal DNA binding [GO:0031492]; structural constituent of chromatin [GO:0030527]	
A0A4X1SY39_PIG	Profilin		2.0095	0.011288	actin cytoskeleton organization [GO:0030036]; positive regulation of actin filament bundle assembly [GO:0032233]; regulation of actin filament polymerization [GO:0030833]	Cytoplasm [GO:0005737]	Actin binding [GO:0003779]	
A0A480VK41_PIG	Calcium-binding mitochondrial carrier protein SCaMC-3		13.907	1.2381E−13	ADP transport [GO:0015866]	Mitochondrial inner membrane [GO:0005743]	ATP transmembrane transporter activity [GO:0005347]; calcium ion binding [GO:0005509]	
A0A5G2R862_PIG	Phospholipase B-like (EC 3.1.1.-)	PLBD2	8.8088	4.6594E−09	Lipid catabolic process [GO:0016042]		phospholipase activity [GO:0004620]	
A0A5G2QHQ3_PIG	Cadherin related family member 2	CDHR2	6.9728	2.6614E−07	Homophilic cell adhesion via plasma membrane adhesion molecules [GO:0007156]	Plasma membrane [GO:0005886]	Calcium ion binding [GO:0005509]	
A0A4X1U5P4_PIG	Maspardin	SPG21	5.4804	6.6162E−06		Cytosol [GO:0005829]; trans-Golgi network transport vesicle [GO:0030140]	CD4 receptor binding [GO:0042609]	
F1SHK2_PIG	MAP3K12 binding inhibitory protein 1	MAP3K12	3.5414	0.00047907				
A0A287B182_PIG	transmembrane protein 263		2.5168	0.0039686				
STMN1_PIG	Stathmin	STMN1	4.9505	2.1013E−05	Microtubule depolymerization [GO:0007019]; neuron projection development [GO:0031175]; regulation of microtubule polymerization or depolymerization [GO:0031110]	Cytoplasm [GO:0005737]; microtubule [GO:0005874]; neuron projection [GO:0043005]	Tubulin binding [GO:0015631]	
A0A5K1U8K2_PIG	Nuclear receptor coactivator 7	NCOA7	10.369	2.5625E−10				

Protein-protein interaction network analysis of piNSC differentiation

To elucidate the molecular mechanisms governing the differentiation of piNSCs into neuronal and glial lineages (piNSCs-NGs), a protein-protein interaction (PPI) network analysis was integrated with pathway enrichment analysis. This analysis focused on the 19 proteins identified as significantly upregulated in differentiated cells by SWATH-MS (INA, KCNMB3, ENSGGOG00000001927, ANXA4, EXOSC3, HIST1H1D, a J domain-containing protein, ZFPL1, NCOR2, a H15 domain-containing protein, PFN1, SLC25A23, a phospholipase B-like protein, CDHR2, SPG21, MBIP, TMEM263, STMN1, and NCOA7), and six established neural markers: MAP2, beta-3 tubulin (TUBB3; ENSG00000258947), SYP, TH, GFAP, and OLIG1. The resulting PPI network (Fig. 7) reveals distinct interaction patterns. Several of the upregulated proteins (EXOSC3, HIST1H1D, NCOR2, PFN1, MBIP, STMN1, and NCOA7) exhibit multiple connections within the network, including interactions with the neural markers (MAP2, GFAP, TUBB3, TH, and SYP). This suggests that these proteins are integrated into established neuronal and glial differentiation pathways. In contrast, other upregulated proteins (INA, ANXA4, CDHR2, SPG21, and OLIG1) show fewer connections, potentially indicating more specialized roles or involvement in less well-characterized pathways.

Figure 7 Protein-protein interaction (PPI) network of upregulated proteins and neural markers in piNSC differentiation.

The network shows interactions among 19 proteins upregulated in piNSCs-NGs (red circles), six neural markers (light blue circles; MAP2, TUBB3, SYP, TH, GFAP, OLIG1), and other interacting proteins (grey nodes). Nodes: proteins; edges: interactions. Network generated using STITCH (accessed July 28, 2024) based on experimental data, text mining, and curated databases. Edge thickness: interaction confidence. Green edges: activation; red edges: inhibition. Node size: degree of connectivity. Key clusters include exosome components and HDAC2-associated proteins.

Specific interactions within the network highlight potential regulatory mechanisms. PFN1 (profilin 1) and STMN1 show strong connections to ACTB (actin) and GFAP, consistent with their known roles in cytoskeletal dynamics, a process crucial for neuronal and glial differentiation. Interactions between MAP2 (a microtubule-associated protein) and TH (tyrosine hydroxylase) are observed, supporting a potential role for these proteins in dopaminergic neuron differentiation. The network also reveals a prominent cluster of exosome-related proteins, including EXOSC3 and its interacting partners (EXOSC1-9), suggesting a role for RNA metabolism and post-transcriptional regulation in differentiation. The presence of HDAC2 and other transcriptional/epigenetic regulators highlights the importance of chromatin remodeling and gene expression control.

Pathway enrichment analysis (Table S8) corroborated the network findings, revealing significant enrichment of pathways related to RNA processing and degradation. Specifically, the analysis highlighted pathways involving the cellular RNA degradation complex (also known as the RNA exosome), exoribonucleases, and targeted mRNA destabilization. This destabilization is mediated by RNA-binding proteins, such as TTP/ZFP36 and KSRP, which bind to target mRNAs and promote their degradation. These pathways are critical for regulating mRNA turnover during neuronal and glial cell maturation. The enriched pathways are functionally linked to several proteins in the PPI network (e.g., the exosome components), further supporting the importance of RNA metabolism in piNSC differentiation. This integrated network and pathway analysis provides a systems-level view of the molecular mechanisms governing piNSC differentiation, identifying key proteins and pathways that are promising targets for future research in neural development and regenerative medicine.

Discussion

This study provides a comprehensive proteomic characterization of the molecular landscape associated with the reprogramming of PTFs into piNSCs and their subsequent differentiation into piNSCs-NGs. By employing label-free nano-LC-MS/MS coupled with targeted SWATH-MS validation, we generated a robust dataset revealing the dynamic protein expression changes distinct protein expression profiles associated with these crucial cellular fate states in a large animal model. Our findings offer significant insights into the molecular mechanisms governing porcine neural induction and differentiation, identifying key protein candidates and pathways with potential utility for regenerative medicine strategies.

The initial characterization confirmed the successful generation and neural commitment of the VSMUi002-B and VSMUi002-E piNSC lines. These cells displayed characteristic neuroepithelial morphology and robustly expressed key NSC markers (PAX6, SOX2, NES, VIM) along with the absence of the pluripotency marker OCT4, indicating a committed NSC state consistent with successful direct reprogramming. Upon induction, these piNSCs demonstrated potent differentiation capacity, generating diverse cell populations expressing markers for immature (TUJ1) and mature neurons (MAP2), synaptic components (SYP), dopaminergic neurons (TH), and astrocytes (GFAP). This validates the suitability of these specific piNSC lines as a reliable in vitro system for studying porcine neural development and for potential use in modeling neurological diseases.

Analysis of the global proteome revealed distinct protein expression signatures for PTFs, piNSCs, and piNSCs-NGs, as visualized through heatmap clustering and quantitatively confirmed by multivariate statistical analyses (PLS-DA, HCA). This clear separation underscores the profound proteomic remodeling inherent to both reprogramming and differentiation.

The observed clustering proximity between the two independent piNSC lines, and similarly between their differentiated progeny, supports the reproducibility of achieving these cellular states, despite detectable line-specific variations. Initial pathway enrichment analyses of the global proteomic data highlighted terms related to “ribosome”, “regulation of mRNA metabolic process”, and “mRNA metabolic process”. This finding suggests the crucial role of the protein synthesis machinery and its regulation in establishing these cell fate states. Ribosomes are fundamental for translating mRNA into the vast array of proteins required for cell proliferation, differentiation, and identity changes (Han et al., 2020). Efficient translation is critical, as disruptions can compromise stem cell function and viability. Beyond their fundamental role in biosynthetic output, ribosomes are increasingly recognized as regulatory hubs influencing specific mRNA translation and stem cell gene expression. This contributes to the precise control needed for stem cell self-renewal and differentiation, potentially impacting epigenetic modifications (Gerst, 2018; Han et al., 2020). Therefore, the prominent enrichment of ribosomal pathways in our dataset strongly reflects the substantial biosynthetic demands required for the extensive cellular remodeling to establish the reprogrammed and differentiated states.

Complementing the demands on protein synthesis, cellular metabolism plays a critical role by providing essential energy and building blocks for cellular reprogramming and differentiation. Furthermore, metabolites serve as critical substrates and cofactors for epigenetic modifying enzymes, reflecting the well-established interconnection between metabolic alterations and epigenetic reprogramming (Miyazawa & Aulehla, 2018). This metabolism-epigenetics coupling is exemplified during mammalian corticogenesis, where temporal changes occur in NSC metabolic gene expression (Okamoto et al., 2016) and the epigenetic landscape (Albert, 2023). Given that metabolites directly act as substrates/cofactors for epigenetic modifiers (Miyazawa & Aulehla, 2018), these coordinated dynamics strongly suggest that co-regulation guides neurodevelopment. Therefore, the enrichment of pathways related to mRNA metabolic processes, observed alongside ribosomal pathways in our initial analysis, highlights the fundamental importance of regulating mRNA stability and processing. This regulatory layer appears crucial, in addition to the control of protein synthesis itself, for the establishment of the reprogrammed piNSCs and subsequent neural state in this porcine system.

A comparative proteomic analysis between PTFs and piNSCs revealed significant molecular restructuring consistent with a profound cell fate change that establishes an NSC identity while bypassing pluripotency. While a core proteome of 521 proteins persisted across PTFs and piNSCs, likely representing essential cellular functions, the large number of unique and differentially expressed proteins signifies a major identity shift. This shift involves the expected activation of programs associated with NSC identity and function, alongside the concurrent silencing of programs defining the original mesenchymal fibroblast state. The greater proteomic similarity observed between the two independent piNSC lines compared to the parental PTFs further indicates their convergence towards a defined and reproducible NSC state. To elucidate key molecular signatures of this reprogrammed state, targeted SWATH-MS quantification pinpointed eight proteins with robustly validated differential abundance when comparing piNSCs to their parental PTFs; these included six upregulated proteins (STMN1, NEFL, ACO2, ETFB, ALDOB, and TKT) and two downregulated proteins (ARPC5 and LIMA1).

Among these differentially abundant proteins, STMN1 was significantly upregulated in the resulting piNSCs, as confirmed by SWATH-MS quantification. This upregulation aligns strongly with published studies indicating a crucial role for STMN1 in the acquisition and maintenance of a proliferative NSC or progenitor state, consistent with its established function in facilitating cell division (Boekhoorn et al., 2014; Liu et al., 2023). STMN1 is a key regulator of microtubule (MT) dynamics, essential for cell cycle progression and mitosis (Liu et al., 2023). Its expression is notably high in the nervous system, particularly during developmental periods characterized by intense neurogenesis (Gagliardi et al., 2022) and in adult neurogenic areas (Boekhoorn et al., 2014). Moreover, STMN1 expression is specifically associated with proliferative neural precursors, and its presence is required for normal proliferation in adult neurogenesis (Boekhoorn et al., 2014). Therefore, the observed significant increase in STMN1 abundance in piNSCs serves as a robust proteomic signature validating the acquisition of a proliferative neural stem/progenitor state.

The commitment to a neural lineage is further solidified by a significant increase in the level of neurofilament light chain (NEFL). As a core component of the neuronal cytoskeleton (Yuan et al., 2012), its upregulation clearly indicates successful commitment to a neural lineage (Teo, Tan & Yim, 2014). Given that NEFL protein is characteristically abundant in differentiated neurons (Sainio et al., 2022; Coppens et al., 2023; Bavato et al., 2024), its elevated level in the piNSC population, which primarily exhibits progenitor markers, requires interpretation. While direct reprogramming aims to establish a specific progenitor state, heterogeneity within the resulting cell population is common. Therefore, a plausible explanation for the elevated NEFL protein level is the presence of a subpopulation within the piNSC culture that has already initiated spontaneous neuronal differentiation, even under conditions favouring self-renewal. It is also noteworthy that a study has reported that NEFL gene expression can be significantly increased in NPCs compared to precursor cells during certain neural induction protocols (Huat et al., 2014). Although NEFL protein accumulation is strongly associated with later neuronal maturation, this report suggests that transcriptional activation might occur at the progenitor stage in specific contexts. Thus, the NEFL protein detected in our piNSCs could represent a combination of signals: both from early spontaneously differentiating cells and potentially from an intrinsic upregulation reflecting early neural commitment within the progenitor state itself, consistent with these gene expression findings (Huat et al., 2014). Further investigation, potentially using single-cell resolution techniques to assess both transcript and protein levels, would be valuable to clarify the relative contributions of these sources. Regardless of the precise underlying mechanism responsible for its expression at this stage, the clear upregulation of NEFL protein compared to parental PTFs clearly indicates the successful acquisition of a neural lineage character and differentiates the reprogrammed state from the original somatic origin.

The significant downregulation of LIMA1 (EPLIN) in piNSCs compared to PTFs is a critical step in deconstructing the original fibroblast identity. LIMA1 is a crucial cytoskeletal protein essential for organizing the actin cytoskeleton and stabilizing adherens junctions by linking cadherin-catenin complexes to F-actin (Abe & Takeichi, 2008; Wang et al., 2023). It also functions as a potent stabilizer of the actin filament network, bundling filaments into prominent stress fibers (Maul et al., 2003; Duethorn et al., 2022). These stress fibers, when stabilized by high LIMA1 expression, typically define the rigid, adherent morphology of epithelial cells, in contrast to mesenchymal cells like fibroblasts, where LIMA1 expression is often reduced or lost (Maul et al., 2003; Zhang et al., 2011). While high LIMA1 expression is known to stabilize the cortical actin required to maintain certain stem cell states (e.g., in pluripotent stem cells) (Duethorn et al., 2022), its downregulation here is particularly revealing. It signifies that the molecular requirements for initiating a direct lineage switch favor plasticity over stability. Therefore, the observed downregulation of LIMA1 represents a targeted demolition of the fibroblast’s structural framework. By reducing the levels of this key stabilizing protein, the cell actively promotes the disassembly of its stress fiber network, which typically leads to a more dynamic and less adherent state, often associated with epithelial-to-mesenchymal transition (EMT) (Zhang et al., 2011). Notably, the specific functions of LIMA1 within canonical neural stem or progenitor populations remain poorly defined, underscoring the novelty of our findings in uncovering its pivotal role in the establishment of a neural identity. We propose that the downregulation of LIMA1 is not merely a consequence of reprogramming but a rate-limiting step that is essential for overcoming the physical barriers of the original cell fate.

The downregulation of key enzymes reflects the active suppression of the fibroblast phenotype while cells adapt to the demands of the new piNSC state. Reduced expression of actin-related protein 2/3 complex subunit 5 (ARPC5) exemplifies the structural remodeling away from the fibroblast phenotype. ARPC5 is an essential subunit of the Arp2/3 complex, the major nucleator of branched dendritic actin networks required for lamellipodia formation and mesenchymal cell motility (Pollard, 2007; Suraneni et al., 2012). Fibroblasts heavily rely on these structures for migration (Suraneni et al., 2012). Therefore, the observed downregulation of ARPC5 in piNSCs compared to PTFs signifies a reduced capacity for generating these networks, consistent with the loss of the motile fibroblast phenotype and the adoption of a more stationary, neuroepithelial morphology characteristic of NSCs (Han et al., 2012; Nunes-Santos et al., 2023). This ARPC5 downregulation likely reflects the deconstructing of the fibroblast’s migratory machinery as the cell adopts an NSC state where different actin structures predominate (Suraneni et al., 2012; Nunes-Santos et al., 2023).

Beyond the cytoskeletal restructuring evident from STMN1, NEFL, LIMA1, and ARPC5 changes, profound metabolic reprogramming was observed. This was evidenced by the coordinated upregulation of key metabolic enzymes and the concurrent downregulation of proteins central to the original fibroblast’s structural identity.

The observed upregulation of transketolase (TKT) strongly supports the metabolic shift towards a proliferative state by connecting the non-oxidative PPP back to glycolysis and thereby facilitating the production of essential precursors (Patra & Hay, 2014; Stincone et al., 2015). Specifically, TKT activity plays a vital role in generating both Ribose-5-phosphate (R5P) for nucleotide biosynthesis and NADPH for reductive biosynthesis and defense against oxidative stress, processes crucial for the proliferation and maintenance of stem cell states like piNSCs (Folmes et al., 2012; Jin & Zhou, 2019). The interconnectedness of glycolysis and the PPP, with TKT acting as a key link, allows cells to balance their energy production and biosynthetic needs (Folmes et al., 2012; Patra & Hay, 2014).

Moreover, the upregulation of ALDOB in piNSCs suggests an enhanced glycolytic activity supporting their energy metabolism. ALDOB is one of three vertebrate aldolase isoenzymes involved in glycolysis (Chang et al., 2018). Increased glycolytic flux is a well-documented metabolic adaptation during cellular reprogramming toward pluripotency, as it helps meet the high demands for energy and biosynthetic precursors (Folmes et al., 2013; Bu et al., 2018). This shift likely involves isoform switching, a common feature during cell fate transitions that ensures optimal enzyme activity for regulating pluripotency (Chepelev & Chen, 2013). Therefore, the upregulation of ALDOB in piNSCs may reflect this type of isoform modulation, facilitating efficient glycolysis to sustain their proliferative and metabolic requirements.

ACO2 and ETFB are key mitochondrial enzymes in cellular metabolism. ACO2 is essential for the TCA cycle, the central hub of aerobic energy generation (Zhu et al., 2023). In contrast, ETFB, as part of the electron transfer flavoprotein (ETF) complex, links fatty acid oxidation (FAO) and amino acid catabolism to the electron transport chain (Henriques et al., 2021). In humans, fibroblasts often rely heavily on mitochondrial respiration, whereas progenitor cells like NSCs typically favor aerobic glycolysis and the pentose phosphate pathway (Chen & Chan, 2017). Interestingly, our piNSCs, derived from pigtail fibroblasts, showed upregulation of ACO2 and ETFB. This indicates a robust metabolic transition toward the TCA cycle, FAO, and oxidative phosphorylation (Henriques et al., 2021; Zhu et al., 2023). This divergence might arise from species-specific differences or unique reprogramming pathways in porcine cells compared to human models. Another plausible reason is that the piNSCs may retain an epigenetic memory from their fibroblast origin, which predisposes them to an oxidative metabolism (Folmes et al., 2013; Polo et al., 2010). The increase in ACO2 and ETFB expression in piNSCs suggests a shift toward mitochondrial respiration, which differs from the canonical glycolytic metabolism of native NSCs in human and mouse models. Further investigation is needed to understand the functional implications of this shift and its impact on piNSC biology.

In summary, the differential expression of these eight proteins highlights key molecular events that characterize PTF-to-piNSC reprogramming. The upregulation of NEFL, STMN1, TKT, and ALDOB signifies a concerted effort to: (1) establish a neural-specific cytoskeletal identity, marked by NEFL induction and supported by the dynamic regulation of microtubules (STMN1); (2) enable the high proliferation rate characteristic of piNSCs through cell cycle control (STMN1) and metabolic support (TKT and ALDOB); and (3) adapt cellular metabolism via the PPP (TKT) and glycolysis (ALDOB) to fuel biosynthesis and manage redox stress inherent to this demanding transition. Concurrently, the downregulation of ARPC5 and LIMA1 underscores the critical need to dismantle the original fibroblast identity. Reduced ARPC5 signifies cytoskeletal remodeling away from fibroblast motility, while LIMA1 downregulation represents a targeted demolition of the fibroblast’s structural framework, promoting a more plastic state. The coordinated upregulation of ACO2 and ETFB reflects mitochondrial respiration process that may be influenced by epigenetic memory retained from the fibroblast origin. The co-occurrence of both glycolytic and oxidative signatures in our piNSCs highlights a flexible, hybrid metabolic state that warrants further investigation to understand its functional implications.

A comparison of the proteomes from the undifferentiated and differentiated states (piNSC to piNSCs-NGs), our proteomic analysis revealed both continuity and significant change in the cellular protein landscape. A core proteome of 426 proteins was maintained across both undifferentiated piNSCs and differentiated piNSCs-NGs, likely representing proteins essential for fundamental neuronal and glial cell viability and housekeeping functions. Crucially, 69 proteins were detected exclusively in both differentiated piNSCs-NGs lines, representing strong candidates for markers or drivers of porcine neural differentiation that merit further investigation. Targeted SWATH-MS quantification then confirmed the significant upregulation of 19 specific proteins in piNSCs-NGs compared to piNSCs (Table 3). This upregulated cohort includes proteins with established roles in neuronal structure and function, such as alpha-internexin (INA), a neuron-specific intermediate filament protein (Yuan et al., 2006), and STMN1, which interestingly was also upregulated during reprogramming and is implicated in microtubule dynamics crucial for processes like neurite outgrowth (Boekhoorn et al., 2014; Liu et al., 2023). A detailed examination of these 19 upregulated proteins provides deeper insights into the molecular orchestration of porcine neural differentiation. The protein-protein interaction network analysis integrated these upregulated proteins with established neural markers (MAP2, TUBB3, SYP, TH, GFAP), revealing potential functional relationships and further supporting the successful establishment of neuronal and glial fates.

The differentiation process necessitates dramatic morphological changes, reflected by the upregulation of key cytoskeletal regulators. Establishing the complex architectures of neurons and glia requires dynamic remodeling of all three cytoskeletal systems. The upregulation of alpha-internexin (INA), a Class IV neuronal intermediate filament protein, is critical for building the neuron-specific axoskeleton (Chien, Lee & Lu, 1998; Bavato et al., 2024). INA significantly influences neuronal morphogenesis during development, contributing to neurite outgrowth, axonal caliber, and stability (Chien, Mason & Liem, 1996; Chien, Lee & Lu, 1998; Yuan et al., 2006), while also functioning to maintain the structural integrity of mature neuronal processes (Yuan et al., 2006; Coppens et al., 2023). Its upregulation thus likely supports both the initial formation and subsequent stabilization of neuronal structures as differentiation proceeds. Concurrently, increased profilin-1 (PFN1) expression likely reflects the necessity to orchestrate the complex cytoskeletal transformations required for differentiation. PFN1 drives actin-dependent neuronal morphogenesis, including neurite outgrowth and axonal growth powered by its coordination of actin and microtubules (Pinto-Costa et al., 2020; Di Domenico et al., 2021). PFN1 is also essential for glial morphogenesis, including myelination-related processes (Montani et al., 2014), and contributes to the cell adhesion required for network formation (Di Domenico et al., 2021). This upregulation thus corresponds to the high demand for the functions of PFN1 in building both the neuronal and glial components of the differentiating piNSC-NG culture. Building upon its upregulation during reprogramming, STMN1 levels remained significantly elevated during differentiation. STMN1 primarily regulates microtubule dynamics by promoting microtubule disassembly and preventing tubulin polymerization (Rubin & Atweh, 2004; Boekhoorn et al., 2014; Gagliardi et al., 2022). This activity confers essential plasticity to the microtubule cytoskeleton, crucial for dynamic processes such as neurite outgrowth, axon formation, and potentially cell cycle adjustments as cells commit to a differentiated state (Rubin & Atweh, 2004; Boekhoorn et al., 2014; Chauvin & Sobel, 2015). Therefore, the sustained upregulation of STMN1 during differentiation likely reflects an ongoing necessity for this microtubule plasticity, not just for initiating processes but also for their continued elongation and structural remodeling as neurons and glia mature, complementing the later roles of stabilizing proteins like MAP2. This coordinated upregulation of INA, PFN1, and STMN1 underscores that morphogenesis in the differentiated state is a highly dynamic process involving both the establishment and maintenance of neuron-specific filament networks (INA), extensive actin-driven process outgrowth and adhesion coordinated with microtubule dynamics (PFN1), and regulated microtubule destabilization (STMN1).

The upregulated proteome in differentiated cells also signals the development of specialized cellular functions and their essential metabolic support. For instance, increased KCNMB3, a modulator of MaxiK/BK potassium channels (Hu et al., 2004), signals the maturation of neuronal electrical properties, including excitability, neurotransmitter release, and firing patterns (Hu et al., 2004; Singh et al., 2016). Such heightened neuronal activity demands robust mitochondrial function, supported by upregulated SLC25A23 (SCaMC-3). This mitochondrial ATP-Mg/Pi carrier manages energy charge and aids neuronal responses to excitotoxicity (Hoffman et al., 2014), thereby preparing differentiating piNSCs-NGs for the high energetic demands and activity-related stresses of mature neurons.

Cell fate transitions are fundamentally driven by alterations in gene expression programs, evidenced here by the upregulation of key transcriptional and chromatin regulators. Establishing and stabilizing the neuronal and glial phenotypes requires extensive epigenetic and transcriptional reprogramming. The increased level of H1.3 linker histone (encoded by HIST1H1D) points to large-scale chromatin reorganization, as linker histones are crucial for chromatin compaction and accessibility, likely involved in silencing progenitor genes and defining lineage-specific chromatin states (Fyodorov et al., 2017; Di Liegro, Schiera & Di Liegro, 2018). Complementing this architectural role, the increased expression of the coactivator NCOA7 (ERAP140), which enhances the activity of nuclear receptors responsive to developmental cues like retinoic acid (Arai et al., 2008), is necessary for driving the expression of genes required for differentiation and maturation (Olivares, Moreno-Ramos & Haider, 2016). Collectively, these changes in HIST1H1D and NCOA7 illustrate a dynamic interplay controlling the genome, featuring global chromatin restructuring alongside the specific activation of lineage-appropriate genes to establish new cellular identities.

Post-transcriptional control provides another critical regulatory layer in establishing differentiated cell fates, highlighted by the upregulation of EXOSC3 (Exosome Component 3) in our differentiating piNSCs-NGs. As a core component of the RNA exosome complex (Wan et al., 2012; Fasken et al., 2017), EXOSC3 is essential for the 3′-to-5′ processing and degradation of a wide array of RNA molecules, playing vital roles in RNA maturation, quality control, and turnover (Houseley, LaCava & Tollervey, 2006). Its importance in neural development is underscored by the severe neurodevelopmental phenotypes caused by EXOSC3 mutations (Wan et al., 2012). Consequently, its upregulation in the differentiated state likely reflects the heightened demand for remodeling the transcriptome—clearing progenitor-specific mRNAs, processing newly synthesized neuronal/glial RNAs, eliminating aberrant transcripts, and potentially regulating non-coding RNAs involved in fate decisions (Pefanis et al., 2015; Nair, Chung & Basu, 2020). The observed upregulation of EXOSC3 thus underscores that establishing the differentiated state requires not only transcriptional changes but also robust post-transcriptional management involving critical complexes such as the RNA exosome.

The intricate processes of neuronal and glial maturation also necessitate extensive membrane remodeling, intracellular trafficking, and specific cell interactions, reflected by the upregulation of several key proteins. Increased annexin A4 (ANXA4), a Ca2+-dependent phospholipid-binding protein (Wei et al., 2015), likely supports the membrane fusion and trafficking events essential for neurite outgrowth and synaptogenesis (Winkle & Gupton, 2016; Yao et al., 2016; Vicic et al., 2022). Efficient protein and lipid transport through the secretory pathway, crucial for building extensive neural structures, is correspondingly bolstered by the upregulation of ZFPL1, which maintains cis-Golgi integrity (Chiu et al., 2008). The roles of two other upregulated membrane-associated proteins, CDHR2 (cadherin related family member 2) and TMEM263, are less clear in this neural differentiation context. While CDHR2 organizes specific structures in epithelia (Pinette et al., 2019), its function in neural differentiation is unknown but could involve specific cell-cell recognition or adhesion (Cencer, Robinson & Tyska, 2024). TMEM263 is poorly characterized but linked to skeletal development (Mohajeri et al., 2021); its upregulation here suggests a previously unrecognized role in neural differentiation that merits further investigation. Collectively, these protein changes highlight the critical importance of meticulously managing membrane dynamics, intracellular transport, and cellular organization in the assembly of complex neural cells and networks.

Maintaining cellular integrity and integrating signaling pathways in the differentiated state are crucial processes, supported by the upregulation of proteins associated with protein quality control and signaling modulation. The increased abundance of a J domain-containing protein (JDP), an essential Hsp70 cochaperone (Zhang et al., 2023), underscores the need for robust protein folding and quality control machinery (proteostasis) to manage the synthesis and assembly of the new neuronal/glial proteome and prevent aggregation (Lim & Yue, 2015; Giandomenico, Alvarez-Castelao & Schuman, 2022; Abildgaard et al., 2023). Concurrently, the upregulation of MBIP (MAP3K12 binding inhibitory protein 1) potentially integrates signaling and chromatin modification by inhibiting the MAP3K12/JNK pathway and participating in the ATAC histone acetyltransferase complex (Suganuma et al., 2012). Precise regulation of pathways like JNK is likely critical for balancing survival and differentiation signals (Semba et al., 2020). Collectively, the heightened expression of JDPs and MBIP highlights the sophisticated cellular mechanisms ensuring proteome integrity alongside the coordinated integration of signaling pathways with epigenetic programming, both vital for a robust and stable transition to differentiated neuronal or glial states.

The upregulation of CD27 and SPG21 in differentiated piNSC-NGs is noteworthy, given their primary functions are not typically associated with neural differentiation. CD27, a known immune costimulatory receptor (Schürch et al., 2012; Lee et al., 2013), its increased expression here is unexpected, suggesting potential uncharacterized roles in neural cell interactions or phenomena specific to the in vitro iNSC system. SPG21 (Maspardin), essential for long-term neuronal health and mutated in hereditary spastic paraplegia (Simpson et al., 2003; Davenport et al., 2016), is implicated in endosomal trafficking (Simpson et al., 2003; Davenport et al., 2016); its upregulation may therefore signify an increased investment in mechanisms supporting neuronal maturation and maintenance. These findings for CD27 and SPG21 in our porcine iNSC system invite further investigation into their context-specific roles.

Collectively, the upregulation of these 19 proteins in differentiated piNSCs—spanning cytoskeletal regulators (INA, PFN1, STMN1), factors for functional maturation (KCNMB3, SLC25A23), transcriptional and post-transcriptional machinery (HIST1H1D, NCOA7, EXOSC3), components for membrane dynamics and protein homeostasis (ANXA4, ZFPL1, JDP, MBIP, PLB-like protein—assuming PLB-like is among the 19), and proteins with less defined neural roles (CDHR2, TMEM263, CD27, SPG21)—paints a picture of a highly orchestrated program. This program not only builds specialized neuronal and glial structures but also establishes their functional capabilities and underlying regulatory networks.

Taken together, the comparison between piNSCs and piNSCs-NGs reveals a distinct wave of profound proteomic changes that characterize the emergence of mature neural phenotypes. Dynamic cytoskeletal remodeling, orchestrated by proteins like INA, PFN1, and STMN1, remains paramount for shaping neuronal and glial morphologies. Beyond structural assembly, our data highlight the critical importance of post-transcriptional gene regulation, particularly RNA processing via the RNA exosome complex (evidenced by EXOSC3), as a key control layer.

Methodological strengths and limitations

Methodologically, the strengths of this study include the use of two independently derived and validated piNSC lines, biological replicates (n = 3 per group) to assess variability, the powerful combination of unbiased label-free discovery proteomics with targeted SWATH-MS for quantitative validation, and the integration of proteomic data with prior cell characterization and downstream bioinformatics analyses. The porcine model itself is another key strength, offering greater physiological relevance to humans than rodent models for translational neurological studies and for modeling disease processes relevant to both human and veterinary neurology. Certain limitations, however, should be acknowledged. The primary goal of this study was to establish the proteomic signatures of stable cellular states. Therefore, our analysis represents a “snapshot” at specific passages (p20 for piNSCs and p20+14 days for piNSCs-NGs) and does not capture the transient dynamics during the reprogramming or differentiation processes, which would require dedicated time-course studies. While SWATH-MS improves quantification, the initial label-free discovery phase can exhibit variability and may have limitations in detecting lower-abundance proteins.

Furthermore, this study was intentionally focused on establishing the foundational proteome-level abundance changes in this large animal model. Consequently, a dedicated analysis of PTMs like phosphorylation, while known to be critical regulators of neural development, was beyond the scope of this initial investigation as it requires distinct experimental strategies (e.g., phosphopeptide enrichment) and data analysis pipelines. Investigating the PTM landscape therefore represents an important avenue for future work. Finally, while bioinformatic tools like PPI networks provide valuable hypotheses, these are in silico predictions that require experimental and functional validation (e.g., through gene knockdown or overexpression studies). Similarly, direct extrapolation of findings from this porcine system to human biology requires further confirmation.

Significance and implications

These findings hold significant implications, further establishing the pig as a valuable large animal model for investigating fundamental aspects of neural development and for the crucial preclinical evaluation of regenerative therapies. The identified proteins and pathways associated with reprogramming (e.g., ARPC5-mediated cytoskeletal shifts, metabolic adjustments involving ACO2, ETFB, and TKT) and differentiation (e.g., RNA exosome components like EXOSC3, specific cytoskeletal regulators like STMN1 and INA, and signaling modulators like MBIP) represent potential mechanistic targets (or nodes) for optimizing neural induction protocols or directing specific neuronal/glial fates. Furthermore, this proteomic dataset provides a rich resource for comparative studies, enabling investigations into conserved vs. species-specific mechanisms of neurogenesis and cellular reprogramming across diverse mammalian lineages.

Future directions

Based on these findings, future directions should prioritize rigorous functional validation (e.g., using CRISPR-Cas9 gene editing or siRNA knockdown in the piNSC model) of key identified regulators, particularly those implicated in cytoskeletal dynamics (ARPC5, LIMA1, STMN1, PFN1) and RNA metabolism (EXOSC3, NCOR2, NCOA7). Capturing the temporal dynamics underlying these changes in cell state through time-course proteomic analyses will be essential, directly addressing the snapshot limitation of the current study. This should include not only comprehensive proteomic profiling but also validation of key protein expression patterns at multiple time points using methods like Western blot and qPCR. Integrating these proteomic data with transcriptomic and epigenomic profiling would provide a multi-layered, systems-level view of the regulatory networks at play. Further investigation into the precise metabolic states and fluxes that occur during reprogramming and differentiation, potentially using metabolic tracing techniques, is also warranted. Addressing another limitation of this study, exploring the landscape of PTMs, especially phosphorylation events impacting cytoskeletal regulators and signaling molecules, would add a critical layer of regulatory understanding. Finally, rigorous functional assessment of the generated piNSCs-NGs, including detailed electrophysiology and crucially, in vivo transplantation studies within porcine models, is needed to fully ascertain their maturity and translational potential, bridging the gap between in vitro findings and potential clinical application.

Conclusions

In conclusion, this study successfully delineates the dynamic proteomic landscapes accompanying porcine fibroblast reprogramming into NSCs (piNSCs) and their subsequent neuronal/glial differentiation. We identify distinct proteomic signatures, core maintained proteomes, and sets of significantly altered proteins associated with each transition. Key biological processes implicated include profound cytoskeletal remodeling, significant metabolic reprogramming towards a progenitor-like state during piNSC establishment, and intricate post-transcriptional control via RNA metabolism during differentiation. These data not only enhance our fundamental understanding of neural cell fate determination in a large animal model relevant to human physiology but also provide a valuable proteomic resource and highlight candidate proteins and pathways for future functional studies and potential therapeutic targeting in regenerative neuroscience.

Supplemental Information

Supplemental Information 1 Total Proteins Identified by DDA of PTFs, piNSCs, and piNSCs-NGs using LC MS/MS.

Supplemental Information 2 GO and KEGG Pathway Enrichment Analysis of the Total Identified Proteome.

Supplemental Information 3 Differentially Expressed Proteins Between PTFs and piNSC Lines.

Supplemental Information 4 Differentially expressed neural lineage-related proteins.

Supplemental Information 5 The enrichment pathway analysis fron reprogramming of PTFs to piNSCs, a protein-protein interaction (PPI) network.

Supplemental Information 6 The 28 significant proteins across all four groups and 22 significant proteins between piNSC and NG groups using One-way ANOVA (p < 0.001) and post-hoc Tukey’s test (p < 0.05).

Supplemental Information 7 The 30 significant proteins across the four groups (VSMUi002-B, VSMUi002-E, VSMUi002-B-NGs, and VSMUi002-E-NGs) using One-way ANOVA (p < 0.05) followed by Tukey’s post-hoc test (p < 0.05).

Supplemental Information 8 Enriched pathways from PPI network interaction with upregulated proteins (EXOSC3, HIST1H1D, NCOR2, PFN1, MBIP, STMN1, and NCOA7) and the neural markers (MAP2, GFAP, TUBB3, TH, and SYP).

Supplemental Information 9 Multivariate Analysis of Proteomic Profiles from piNSC Differentiation.

(A) and (B) ANOVA score plot represented the statistical significance of differences in protein abundance of piNSCs lines and differentiated progeny (piNSCs-NGs). (C) Heatmap of Proteins Differentially Expressed Across piNSC and piNSC-NG Groups. (D) Heatmap of Proteins Differentially Expressed Within Differentiated piNSC-NGs.

We thank the Veterinary Science-Frontier Research Facility (VS-FRF) and the Monitoring and Surveillance Center for Zoonotic Diseases in Wildlife and Exotic Animals (MoZWE), along with their technical staff, for providing access to essential laboratory facilities and equipment.

Additional Information and Declarations

Competing Interests

The authors declare that they have no competing interests.

Author Contributions

Sekkarin Ploypetch conceived and designed the experiments, performed the experiments, analyzed the data, prepared figures and/or tables, authored or reviewed drafts of the article, and approved the final draft.

Sataporn Phochantachinda conceived and designed the experiments, performed the experiments, analyzed the data, prepared figures and/or tables, authored or reviewed drafts of the article, and approved the final draft.

Warunya Chakritbudsabong conceived and designed the experiments, performed the experiments, analyzed the data, prepared figures and/or tables, authored or reviewed drafts of the article, and approved the final draft.

Walasinee Sakcamduang conceived and designed the experiments, performed the experiments, analyzed the data, authored or reviewed drafts of the article, and approved the final draft.

Nattarun Chaisilp performed the experiments, authored or reviewed drafts of the article, and approved the final draft.

Somjit Chaiwattanarungruengpaisan performed the experiments, authored or reviewed drafts of the article, and approved the final draft.

Supitcha Pannengpetch performed the experiments, authored or reviewed drafts of the article, and approved the final draft.

Piyada Na Nakorn performed the experiments, authored or reviewed drafts of the article, and approved the final draft.

Tharathip Muangthong performed the experiments, authored or reviewed drafts of the article, and approved the final draft.

Sasitorn Rungarunlert conceived and designed the experiments, performed the experiments, analyzed the data, prepared figures and/or tables, authored or reviewed drafts of the article, and approved the final draft.

Animal Ethics

The following information was supplied relating to ethical approvals (i.e., approving body and any reference numbers):

The Institutional Animal Care and Use Committee (IACUC) at the Faculty of Veterinary Science, Mahidol University, Thailand.

Data Availability

The following information was supplied regarding data availability:

The mass spectrometry proteomics data are available at the ProteomeXchange Consortium via the PRIDE partner repository: PXD063568.

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
