# Peer review of "Proteomic landscape of porcine induced neural stem cell reprogramming and differentiation"

_PeerJ, doi:10.7717/peerj.20120_

## Round 0.1 · original submission · Major Revisions

·

Basic reporting

no comment

Experimental design

no comment

Validity of the findings

no comment

Additional comments

Comments to authors

The study by Ploypetch et al. presents a comprehensive proteomic analysis of porcine induced neural stem cell (piNSC) reprogramming and differentiation, offering valuable insights into the molecular mechanisms governing neural fate transitions in a large animal model. By integrating label-free proteomics with targeted SWATH-MS validation, the authors provide relevant data that advance our understanding. However, several aspects require clarification and additional experimental support to strengthen the manuscript’s conclusions.

1.Why piNSCs at P20 were selected instead of an earlier passage such as P3 to match the fibroblast stage? Whether extended passaging alters piNSC pluripotency or differentiation capacity compared to earlier passages?

2. In the "Proteomic Analysis of PTF Reprogramming to piNSCs" section of the Results, the analysis only elucidate the changes in protein quantity during reprogramming. However, the functional implications of the upregulated and downregulated proteins still need further elaboration through GO enrichment analysis or KEGG analysis presented in the main figures.

3.The study only analyzes fixed time points (P20 and P20+14 days). Should temporal sequence analysis considered to capture earlier or later protein dynamics?

4.Although the potential impact of post-translational modifications (PTMs) is mentioned in both the Introduction and Discussion sections, the Results section lacks corresponding bioinformatics data to support these claims. Please provide supplemental data or explanations to address this gap.

5.How phosphorylation of STMN1 during reprogramming affects its microtubule destabilizing function? Please provide further evidence to confirm the STMN1 phosphorylation and its impact during reprogramming by Western blot or Co-IP analysis.

6.It is expected to provide additional immunofluorescence co-localization images to experimentally validate the crucial protein-protein interactions predicted in Figure 5 and 7.

Reviewer 2 ·

Basic reporting

This study provides a valuable resource for large-animal translational neuroscience with exceptionally high proteomic data quality. Addressing the above revisions will solidify mechanistic claims and broaden impact.
1. ) This study applied a dual-proteomics strategy, identifying a total of 4,094 proteins. The data quality is high, and PCA analysis clearly differentiates the three cellular states, supporting the validity of the conclusions. However, the functional roles of the key targets were not validated. It is essential to select at least 2-3 key regulatory proteins (e.g., STMN1, EXOSC3) and validate their functional necessity in reprogramming or differentiation via knockout or overexpression experiments.
2) The differentiation stage analysis was limited to a single time point (Day 14), lacking early dynamic tracking; this should be supplemented with WB/qPCR validation of the temporal expression patterns of key proteins.

Experimental design

no comment

Validity of the findings

no comment

---

## Round 0.2 · accepted · Accept

Authors have addressed all the concerns of the reviewers

·

Basic reporting

The concerns have been addressed.

Experimental design

-

Validity of the findings

-

External reviews were received for this submission. These reviews were used by the Editor when they made their decision, and can be downloaded below.